# Fermented Gold Kiwi Prevents and Attenuates Chronic Alcohol-Induced Liver Injury in Mice via Suppression of Inflammatory Responses

Jihye Choi [1] , Sangmin Lee [1], Hwal Choi [1], Jeonghyeon Lee [1], Nayong Lee [2], Hyunjeong Oh [2], Hyuckse Kwon [2] and Jungkee Kwon [1,*]

1 Department of Laboratory Animal Medicine, College of Veterinary Medicine, Jeonbuk National University, Iksan-si 54596, Jeollabuk-do, Republic of Korea

2 R & D Team, Food & Supplement Health Claims, Vitech, #602 Giyeon B/D 141 Anjeon-ro, Iseo-myeon, Wanju-gun 55365, Jeollabuk-do, Republic of Korea

* Correspondence: jkwon@jbnu.ac.kr; Tel.: +82-63-850-0951

**Abstract:** Excessive alcohol consumption increases the risk of liver disease and liver-related death. Ninety percent of alcohol consumed is broken down in the liver; excessive consumption destroys liver cells and causes stress. The gold kiwi contains more vitamin C than the green kiwi, and various studies have reported that the gold kiwi boosts digestive health. Fermented gold kiwi (FGK) was made using two lactic acids. It contains many more bioactive compounds than fresh gold kiwi. Mice were first given FGK (50, 125, and 250 mg/kg b.w.) and then given a 5 g/kg alcohol solution (50% *w/v*) for 2 weeks. The results indicate that the FGK promoted hepatic function by significantly decreasing the serum ethanol and aldehyde levels and downgrading the serum TC and TG levels. The FGK attenuated alcohol-induced oxidative stress and improved alcohol metabolism by controlling the ADH and ALDH levels in murine liver tissue. In addition, the FGK significantly reduced the concentration of inflammatory cytokines (TNF-α, IL-1β, and IL-6) in mouse serum and liver tissue. The overexpression of inflammatory mediators (iNOS, COX-2) was also decreased in the FGK groups. This study demonstrates that FGK exerts a protective effect against alcohol-induced liver damage by improving alcohol metabolism and increasing anti-inflammatory activity. This finding suggests that FGK might be developed into a functional food treatment against alcohol-induced liver disease.

**Keywords:** alcoholic liver disease; fermented gold kiwi; inflammation

## 1. Introduction

Alcohol abuse and alcoholism cause severe health and socioeconomic problems worldwide [1]. The mechanisms of alcoholic liver disease (ALD), which is caused by excessive alcohol intake for a long time, involve lipids (steatosis) and inflammation (steatohepatitis) [2]. ALD contributes to cell death, both apoptotic and necrotic, and induces liver injury and disease through various metabolic, inflammatory, and toxic responses [3]. The incidence of ALD is increasing every year because of changes in lifestyle, and ALD has become the second major cause of liver injury [4,5]. ALD is the fourth most common cause of preventable disease-related death in the USA [6].

The activation of inflammatory signaling plays a role in the pathogenesis of ALD [7]. Chronic alcohol intake increases the production of inflammatory mediators such as tumor necrosis factor (TNF)-α and interleukin (IL)-1β and promotes the progression of fatty liver, inflammation, oxidative stress, and fibrogenesis [8]. An increased expression of infectious enzymes such as inducible nitric oxide synthetase (iNOS) and cyclooxygenase-2 (COX-2) can increase the production of inflammatory cytokines [9]. Therefore, preventing the inflammatory response would effectively delay the pathogenesis of alcoholic liver injury.

The kiwi is a nutritious fruit containing vitamin C, dietary fiber, vitamin E, and folic acid. It has recently attracted attention as a functional food ingredient because it contains a wide range of antioxidants, phytonutrients, and enzymes [10]. Several studies have shown that kiwis have antioxidant activity, anticancer activity, and effects that promote digestive health, gastro-protection, and hepato-protection [10–13]. Our previous study demonstrated that fermented gold kiwi (FGK) had a high antioxidant capacity and protected rats from gastric injury. The bioactive compound activity in kiwis increased when they were fermented with two types of lactic acid bacteria (Lactococcus lactis VI-01 and Lactobacillus paracasei VI-02) [12]. However, the effects of FGK against alcohol-induced liver injury in mice has not previously been tested.

Our aim in this study was to investigate the possible hepatoprotective effects of FGK in alcohol-induced liver injury by detecting alcohol metabolism and inflammatory markers in mice. These results elucidate the underlying hepatoprotective mechanisms of FGK against alcohol-induced liver damage in mice.

## 2. Materials and Methods

### 2.1. Chemicals and Reagents

Absolute ethanol (≥99.8%) and ascorbic acid were obtained from Sigma-Aldrich (St. Louis, MO, USA). TNF-α (ab208348), IL-1β (ab242234), and IL-6 (ab222503) ELISA kits; ethanol (ab65343), and aldehyde (ab108306) assay kits; and primary antibodies against alcohol dehydrogenase (ADH, ab108203) and aldehyde dehydrogenase (ALDH, ab108306) were purchased from Abcam (Cambridge, UK). Total cholesterol (TC, BM-CHO-100) and triglyceride (TG, BM-TGR-100) assay kits were obtained from BIOMAX (Seoul, Republic of Korea). Primary antibodies against iNOS (5023S), COX-2 (4842S), TNF-α (19948), IL-1β (12703S), and IL-6 (12912S) and the secondary antibodies goat anti-rabbit IgG (H + L) and goat anti-mouse IgG (H + L) were obtained from Cell Signaling Technology (Beverly, MA, USA).

### 2.2. Sample Preparation

Jeju gold kiwis (*Actinidia chinensis* L.) puree with the peels and seeds removed were obtained from Namuang Foods. The bacteria used to ferment the kiwis, *Lactococcus lactis* VI-01 (KTCT 14351 BP) and *Lactobacillus paracasei* VI-02 (KTCT 14352 BP), were isolated from the Jeju gold kiwi peels. Each strain sample was cultured in MRS broth at 37 °C. The pre-culture solution was mixed with the prepared kiwi puree and cultured at 37 °C for 8 h to prepare the FGK. The cultured fermented product was collected and prepared as a freeze-dried powder [10].

### 2.3. Animal Expreiments and Treatments

Male ICR mice (8-weeks-old) were obtained from Damul (Daejeon, Korea). The animal experiments were performed according to the guidelines prescribed by the Jeonbuk National University Animal Care Committee (Approval Number: JBNU 2022-095). During the experimental period, the mice were given free access to food and water. Mice were monitored once a day for their health status by assessing behavior changes, and their body weight and food intake were checked once a week.

Forty-eight mice were randomly assigned to six groups:

- Group 1: Sham [Normal; only saline treatment] (*n* = 8)
- Group 2: Con [Control; Alcohol 5 g/kg + saline] (*n* = 8)
- Group 3: AA [Positive Control; Alcohol 5 g/kg + ascorbic acid 25 mg/kg] (*n* = 8)
- Group 4: FGK-L [Alcohol 5 g/kg + FGK 50 mg/kg] (*n* = 8)
- Group 5: FGK-M [Alcohol 5 g/kg + FGK 125 mg/kg] (*n* = 8)
- Group 6: FGK-H [Alcohol 5 g/kg + FGK 250 mg/kg] (*n* = 8)

All groups except the normal group (Sham) were given an alcohol solution (50% *w/v*, 5 g/kg BW) after sample administration. Saline was administered to the Sham group and Con group, ascorbic acid was administered to the AA group, and FGK was administered to the experimental groups (FGK-L, M, H), respectively. This process was repeated every day for 14 days. All mice were euthanized 12 h after the last administration, and the blood and liver tissue were separated and stored at −80 °C.

### 2.4. Biochemical Assay Measurement

The serum ethanol and plasma aldehyde concentrations were measured using kits (Abcam, Cambridge, UK). The ethanol concentration was measured using a microplate reader (Synergy 2, BioTek Instrument, Winooski, VT, USA) at 570 nm. The aldehyde concentration was measured using a microplate reader (BioTek Instrument) at 405 nm. Serum TC and TG assays were performed using kits. Serum inflammatory cytokines (TNF-$\alpha$, IL-1$\beta$, and IL-6) were measured using ELISA assay kits (Abcam, Cambridge, UK). The cytokine concentrations were measured using a microplate reader (Synergy 2, BioTek instrument, Winooski, VT, USA) at 450 nm. All biochemical analyses were performed according to the relevant manufacturer protocols.

### 2.5. Histological Analysis

The liver tissue was fixed in 10% formalin for 24 h. All selected samples were fitted into paraffin and cut into 5 μm sections using a microtome. The tissue section samples were deparaffinized using a xylene and graduated alcohol series to water and then stained with hematoxylin and eosin (H and E). After staining, the tissue section samples were observed under an optical microscope (Zeiss, Jena, Germany). Liver steatosis was calculated based on the work of Bedossa et al. [14] and scored as shown in Table 1.

**Table 1.** Liver steatosis damage score.

| Score (Point) | Histological Feature |
| --- | --- |
| 0 | No hepatocytes affected |
| 0.5 | Slight damage (0–5%) |
| 1 | Mild damage (5–20%) |
| 2 | Moderate damage (20–50%) |
| 3 | Severe damage (>50%) |

### 2.6. Western Blotting

Protein samples from the mouse livers were extracted with RIPA buffer (GenDEPOT, Katy, TX, USA) with a 1% protease inhibitor cocktail or phosphatase inhibitor cocktail. The protein concentrations were detected using a BCA assay kit (Thermo Fisher, Middlesex County, MA, USA), and each liver protein sample was electrophoresed on a 10% SDS-PAGE mini gel. After loading, the liver proteins were transferred to polyvinylidene fluoride membranes, and then the membranes were blocked with a blocking buffer (12010947, Bio-Rad, Contra Costa County, CA, USA) for 5 min at room temperature. After blocking, the membranes were incubated with primary antibodies diluted in the blocking buffer at 4 °C overnight. After we washed the membranes three times with PBS-T, we incubated them at room temperature for 1 h with secondary antibodies diluted in a blocking buffer. The membranes were detected using SuperSingal West Dura Extended Duration Substrate (Thermo Fisher) and analyzed using a Chemi-Imager system (Alpha Innotech, San Leandro, CA, USA).

### 2.7. Statistical Analysis

Data are presented as the mean $\pm$ SEM. Significant differences between groups were performed using a one-way ANOVA in GraphPad Prism version 5.0 (GraphPad Software, San Diego, CA, USA), and the differences were considered significant $p < 0.05$ in Tukey's multiple range test.

## 3. Results

### 3.1. Effects of Fermented Gold Kiwi on Alcohol-Induced Liver Injury

As shown in Figure 1a, liver enlargement and macrovesicular steatosis were found in the Con (alcohol-alone) group, compared with the Sham group, and the FGK pretreatment mitigated those changes. The liver index is the mass ratio of liver weight to body weight. As shown in Figure 1b, the Con group showed an increase in liver weight compared with the Sham group. These results indicate that continual alcohol gavage caused liver enlargement in mice [15]. The liver index was significantly decreased by the FGK treatment compared with the Con group ($p < 0.05$): by 7.8%, 8.7%, and 9.1%, respectively, in the FGK-L, -M, and -H groups. Therefore, alcohol influenced the liver weight in mice, and the FGK treatment relieved alcohol-related liver enlargement.

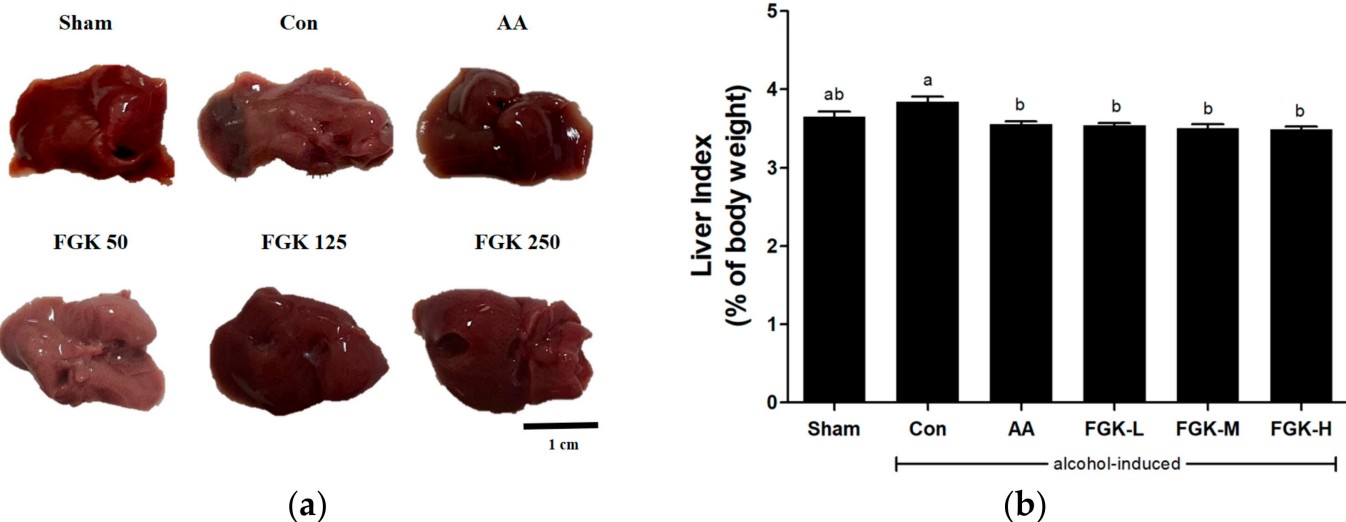

<div align="center">(a)         (b)</div>

**Figure 1.** Effects of fermented gold kiwi (FGK) on the liver index of alcohol-induced mice. (**a**) Gross examination of mouse liver; (**b**) liver index levels. Sham, mice treated with only saline; Con, mice treated with alcohol and saline; AA, mice treated with alcohol and ascorbic acid 25 mg/kg; FGK-L, -M, -H, mice treated with alcohol and FGK at 50, 125 and 250 mg/kg, respectively. Results are expressed as means $\pm$ SEM ($n = 8$). [a,b] Different letters indicate significant differences $p < 0.05$ among the groups, as determined by Tukey's analysis.

To confirm to protective effect of the FGK against hepatic steatosis induced by chronic alcohol consumption, liver tissues were histologically analyzed using H and E staining (Figure 2a). As shown in Figure 2, liver tissues from the alcohol-only mice (Con group) showed signs of fatty liver in more hepatocytes than in the Sham group. The pathological changes seen in the Con group were attenuated by the FGK pretreatment. Furthermore, the liver steatosis score was remarkably higher in the Con group than the Sham group, and the change was significantly decreased by the FGK treatment in a dose-dependent manner (Figure 2b). Overall, the pathology and liver injury results suggest that the FGK treatment protects the liver structure in alcohol-treated mice and inhibits lipid synthesis in the liver.

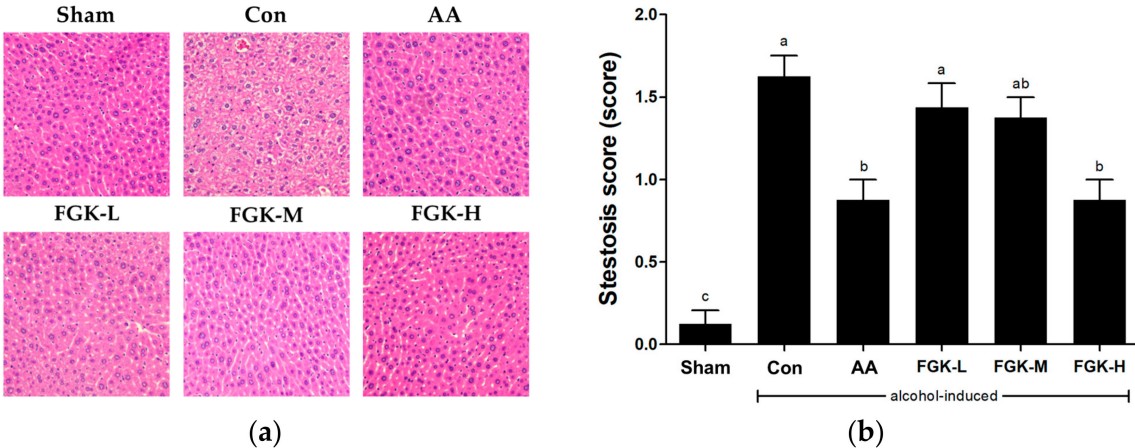

**Figure 2.** Effects of fermented gold kiwi (FGK) on the livers of alcohol-induced mice. (**a**) Histopathological changes; (**b**) steatosis score. Sham, mice treated with only saline; Con, mice treated with alcohol and saline; AA, mice treated with alcohol and ascorbic acid (25 mg/kg); FGK-L, -M, -H, mice treated with alcohol and FGK at 50, 126, and 250 mg/kg, respectively. Results expression as means ± SEM (*n* = 8). [a–c] Different letters indicate significant differences (*p* < 0.05) among the groups, as determined by Tukey's analysis.

### 3.2. Effects of Fermented Gold Kiwi Powder on Alcohol Metabolism

The serum ethanol levels in the mice were measured using an ethanol assay kit to determine whether the FGK could lower the blood ethanol concentration in mice chronically exposed to alcohol (Figure 3a). No serum ethanol was detected in the Sham group (6.5 ± 0.3 nmol), and the Con group (48.8 ± 0.5 nmol) had a high serum ethanol concentration. Interestingly, the serum ethanol concentration decreased significantly, by 9.8% and 13.3 % in the FGK-M and FGK-H groups, respectively, compared with the Con group (*p* < 0.05). In addition, alcohol administration for 2 weeks significantly increased the serum aldehyde concentration in the Con group (Figure 3b), and that increase was reduced in the FGK-L and FGK-M groups, though at an insignificant level (*p* > 0.05). Only the FGK-H group showed a significant decrease (*p* < 0.05) in serum aldehyde concentration, compared with the Con group.

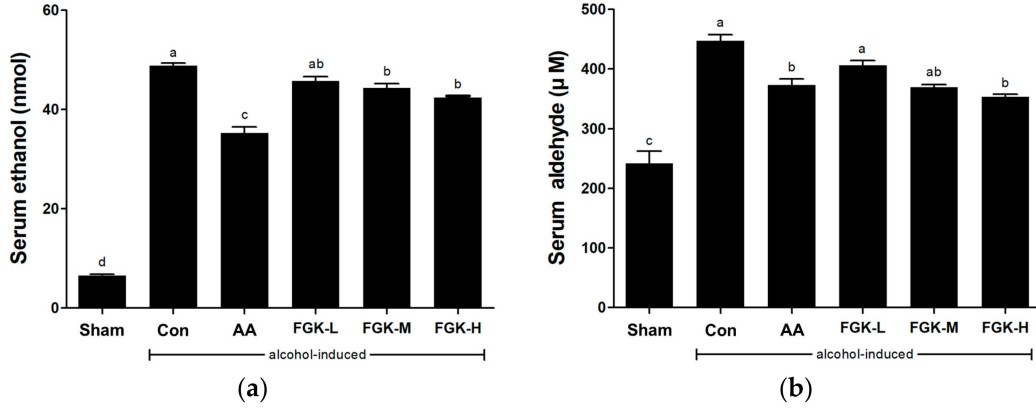

**Figure 3.** Effects of fermented gold kiwi (FGK) on serum alcohol metabolism levels in alcohol-induced mice. (**a**) Ethanol concentration; (**b**) aldehyde concentration. Sham, mice treated with only saline; Con, mice treated with alcohol and saline; AA, mice treated with alcohol and ascorbic acid (25 mg/kg); FGK-L, -M, -H, mice treated with alcohol and FGK at 50, 126, and 250 mg/kg, respectively. Results expression as means ± SEM (*n* = 8). [a–d] Different letters indicate significant differences (*p* < 0.05) among the groups, as determined by Tukey's analysis.

The effects of alcohol administration on hepatic ADH and ALDH activity were examined next (Figure 4). ADH is involved in the liver's major pathway of ethanol metabolism under normal physiological conditions [16]. The protein expression of ADH decreased in the Con group (Figure 4b), but the pretreatment at the FGK-H level protected against that alcohol-induced ADH decrease. Hepatic ALDH responses to alcohol in mice (Figure 4c) showed a decrease in hepatic ALDH in the Con group compared with the Sham group, and the FGK groups showed a significant increase ($p < 0.05$).

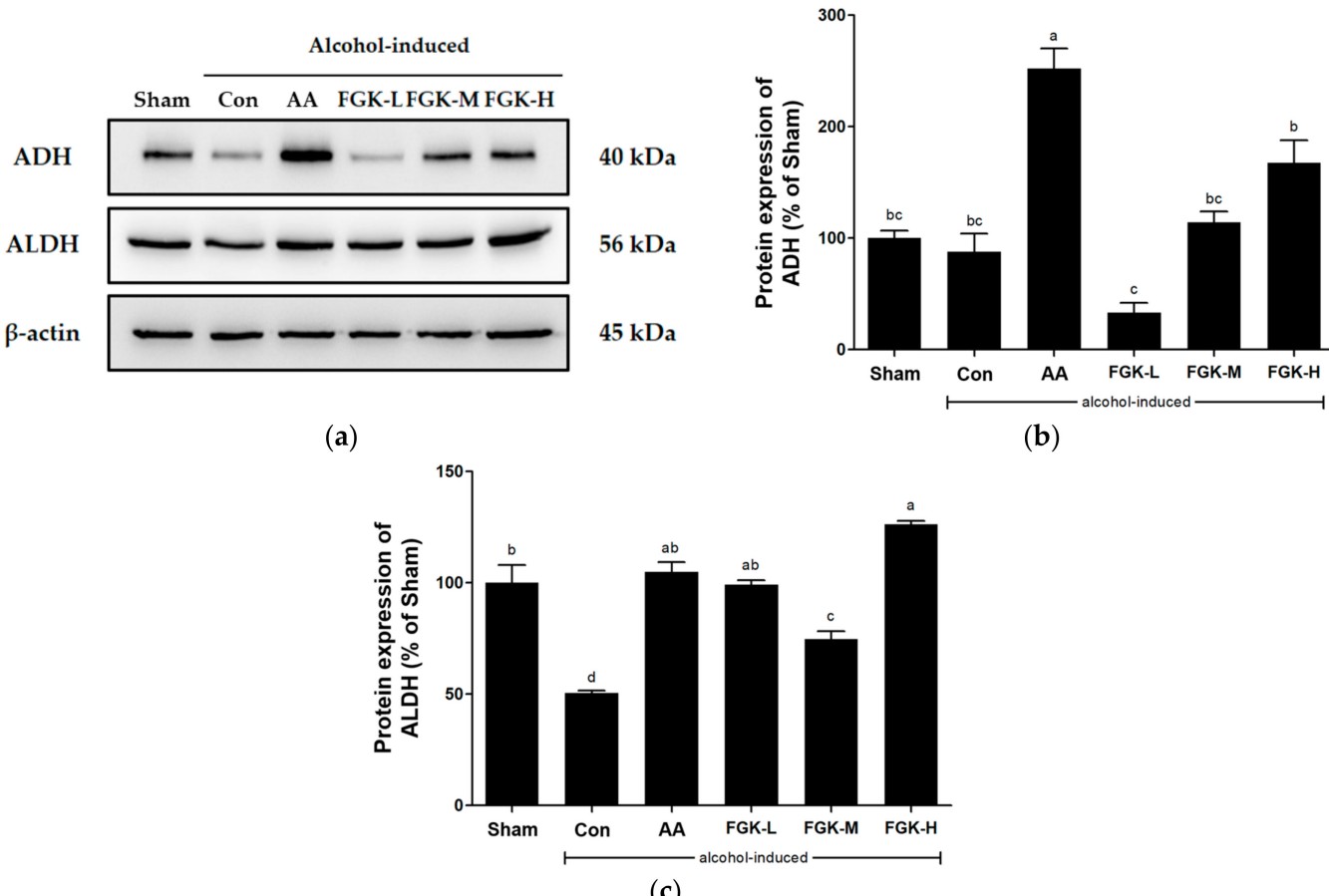

**Figure 4.** Effects of fermented gold kiwi (FGK) on liver alcohol metabolism in alcohol-induced mice. (**a**) Protein expression; (**b**) alcohol dehydrogenase (ADH); (**c**) aldehyde dehydrogenase (ALDH). Sham, mice treated with only saline; Con, mice treated with alcohol and saline; AA, mice treated with alcohol and ascorbic acid (25 mg/kg); FGK-L, -M, -H, mice treated with alcohol and FGK at 50, 126, and 250 mg/kg, respectively. Results expression as means ± SEM (*n* = 8). [a–d] Different letters indicate significant differences ($p < 0.05$) among the groups, as determined by Tukey's analysis.

### 3.3. Effects of Fermented Gold Kiwi Powder on Alcohol-Induced Lipid Profile

The serum TG level increased significantly in the Con group compared with the Sham group (Figure 5a), and the pretreatment with FGK for 2 weeks significantly attenuated those changes, especially in the FGK-H (250 mg/kg) group. In addition, the serum TC level in the Con group was higher than in the Sham group (Figure 5b), and the FGK treatment decreased the levels of serum TC by 12.6%, 40.2%, and 51.9% in the FGK-L, -M, and -H groups, respectively. Recent research reported increased serum TC and TG levels in the alcohol-treated mice [17], and this study confirmed those results and shows that FGK administration attenuated those effects.

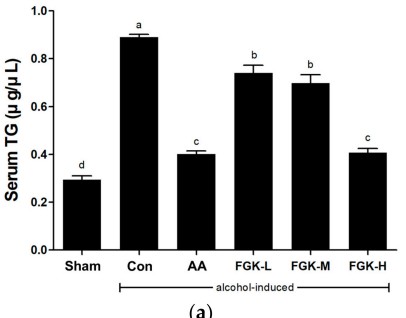 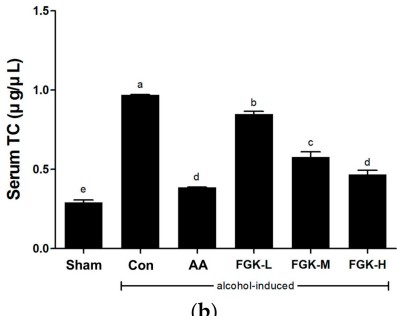

(a)                                (b)

**Figure 5.** Effects of fermented gold kiwi (FGK) on serum lipid profile levels in alcohol-induced mice. (**a**) Triglyceride (TG); (**b**) total cholesterol (TC). Sham, mice treated with only saline; Con, mice treated with alcohol and saline; AA, mice treated with alcohol and ascorbic acid (25 mg/kg); FGK-L, -M, -H, mice treated with alcohol and FGK at 50, 126, and 250 mg/kg, respectively. Results expression as means ± SEM (*n* = 8). [a–e] Different letters indicate significant differences (*p* < 0.05) among the groups, as determined by Tukey's analysis.

### 3.4. Effects of Fermented Gold Kiwi Powder on Alcohol-Induced Inflammatory Mediators

The chronic intake of alcohol activates Kupffer cells and the toll-like receptor (TLR)-4 pathway, which increases the inflammatory cytokines such as TNF-α, IL-1β, and IL-6 and inhibits the secretion of anti-inflammatory cytokines such as IL-10 [18,19]. To evaluate the effects of the FGK on alcohol-induced liver inflammation, we tested the concentration of inflammatory cytokines in serum. As shown as Figure 6, the Con group showed significantly higher cytokine levels than the Sham group, and the FGK pretreatment dose-dependently and significantly inhibited that increase in pro-inflammatory cytokines (TNF-α, IL-1β, and IL-6).

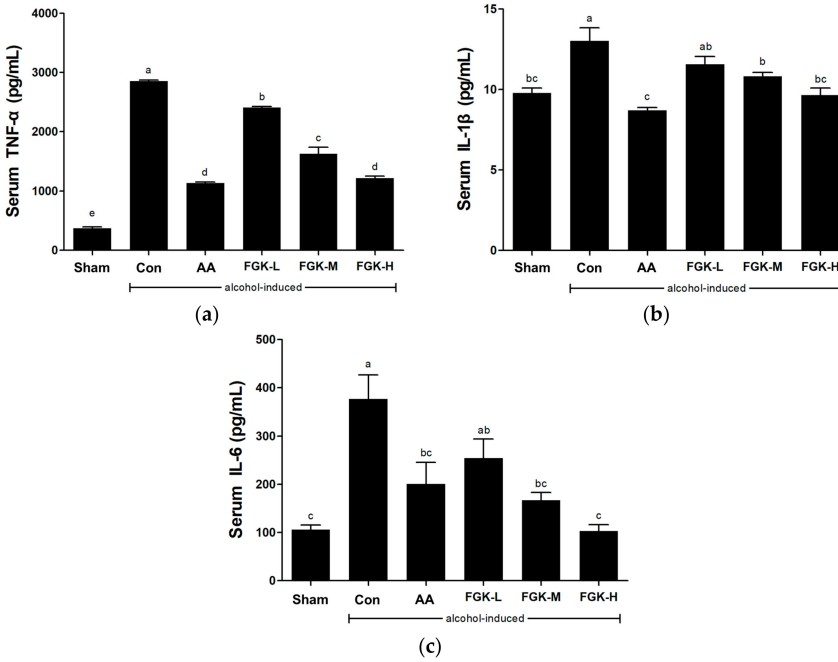

**Figure 6.** Effects of fermented gold kiwi (FGK) on serum cytokines levels in alcohol-induced mice. (**a**) TNF-α, (**b**) IL-1β, and (**c**) IL-6. Sham, mice treated with only saline; Con, mice treated with alcohol and saline; AA, mice treated with alcohol and ascorbic acid (25 mg/kg); FGK-L, -M, -H, mice treated with alcohol and FGK at 50, 126, and 250 mg/kg, respectively. Results expression as means ± SEM (*n* = 8). [a–e] Different letters indicate significant differences (*p* < 0.05) among the groups, as determined by Tukey's analysis.

The expression of inflammation-related factors was confirmed to evaluate the effect of the FGK on alcohol-induced inflammation. As shown in Figure 7a, the levels of inflammatory markers were significantly increased by alcohol (5 g/kg). iNOS and COX-2 are important inflammatory markers for the activation of inflammatory cytokine production, which plays a role in the liver damage caused by alcohol [20]. These results show that iNOS and COX-2 expression increased in the Con group (Figure 7b,c). Furthermore, the overexpression of iNOS and COX in the Con group decreased dose-dependently with the FGK pretreatment. Several studies have shown that pro-inflammatory cytokines such as TNF-$\alpha$ play an important role in the treatment of the alcohol-induced liver injury [20]. We therefore used a Western blot analysis to investigate whether the FGK inhibited pro-inflammatory cytokines. The expression of cytokine markers was upregulated in the Con group mice, and the FGK downregulated the expression of those cytokines in the FGK-H group (Figure 7d–f). Taken together, these results support the ability of the FGK to inhibit inflammatory mediators of alcohol-induced liver injury in mice.

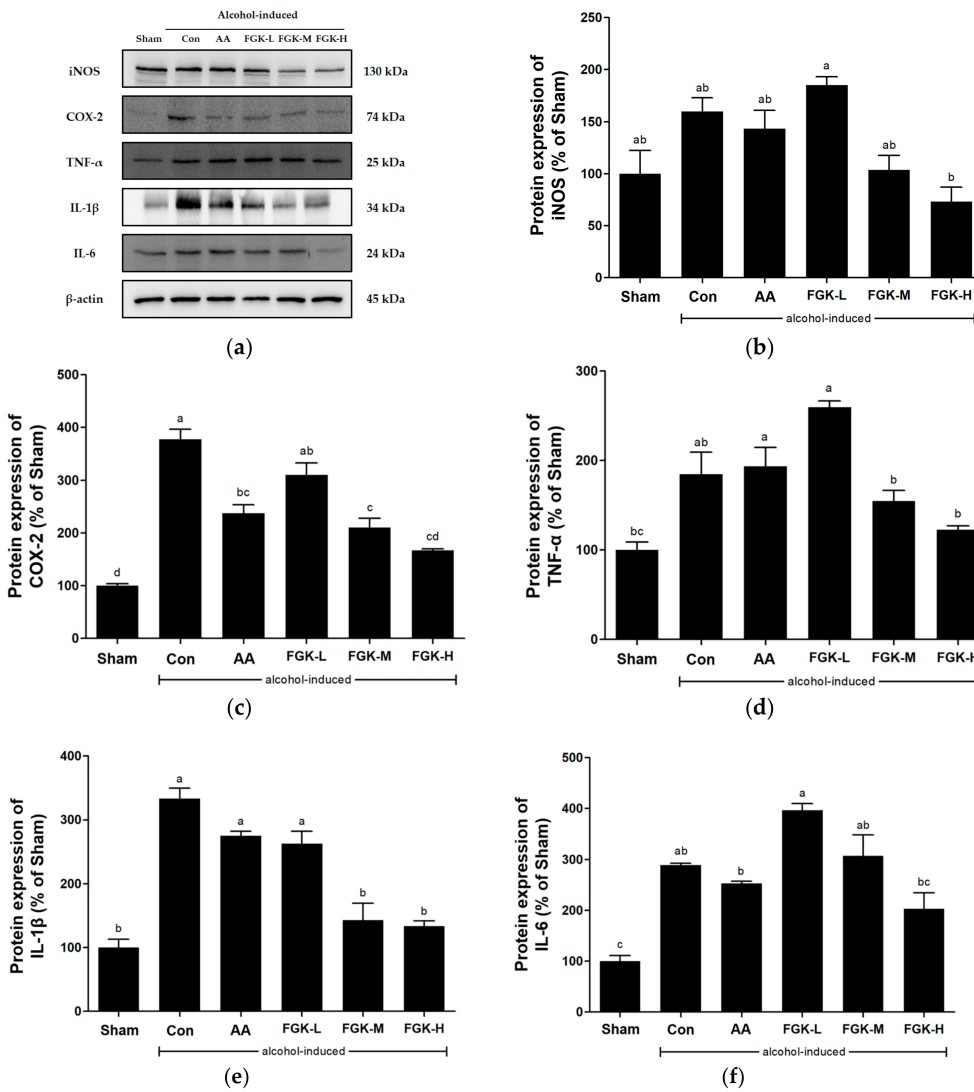

**Figure 7.** Effects of fermented gold kiwi (FGK) on liver inflammatory mediator levels in alcohol-induced mice. (**a**) Protein expression; (**b**) iNOS; (**c**) COX-2; (**d**) TNF-$\alpha$; (**e**) IL-1$\beta$; and (**f**) IL-6. Sham, mice treated with only saline; Con, mice treated with alcohol and saline; AA, mice treated with alcohol and ascorbic acid (25 mg/kg); FGK-L, -M, -H, mice treated with alcohol and FGK at 50, 126, and 250 mg/kg, respectively. Results expression as means $\pm$ SEM ($n = 8$). [a–d] Different letters indicate significant differences ($p < 0.05$) among the groups, as determined by Tukey's analysis.

## 4. Discussion

Recent studies into the kiwi have focused on its bioactive potential. Kiwis have a high vitamin C content and contain many nutrients that help with constipation and gastrointestinal problems [10,20]. Kiwis are rich in nutrients, but as with all seasonal fruits, they have a short shelf life and preserving them causes nutritional changes [21]. Therefore, recent research on biological activity and nutritional characteristic improvement have used fermentation to extend the shelf life of kiwis [12,21,22]. In our previous study, we fermented kiwi puree using two lactic acids (*Lactococcus lactis* VI-01 and *Lactobacillus paracasei* VI-02), both of which were originally isolated from Jeju gold kiwi peels. The FGK increased the levels of organic acids (citric acid, malic acid, quinic acid) and carotenoid content compared with the fresh gold kiwi [12]. In this study, the mice were treated with alcohol (5 g/kg) and the FGK (50, 125 or 250 mg/kg) for 2 weeks to test its protective effect against alcohol-induced liver injury.

ALD is a major cause of chronic liver injury worldwide and eventually leads to fibrosis and cirrhosis [23]. When liver damage is induced by alcohol, an inflammatory reaction occurs in the body [9]. The metabolism of alcohol depends on oxidation and produces metabolic byproducts such as ADH and ALDH [24]. Alcohol is broken down into acetaldehyde by ADH, acetate by ALDH, and carbon dioxide and water by the tricarboxylic acid cycle. Therefore, alcohol-induced liver damage is associated with ADH and ALDH activity [25,26]. In addition, ethanol interacts with aldehyde catalysts in the body to produce large amounts of ROS, which induces liver injury [26]. Our results in this study show that the FGK had protective effects against alcohol-induced liver injury in mice by attenuating the ethanol and aldehyde concentrations in serum (Figure 3). In addition, the FGK increased the expression of ADH and ALDH in the liver tissue (Figure 4), which suggests that the FGK has positive effects on alcohol metabolism.

Fat accumulation in the liver is the earliest and most common reaction to excessive alcohol consumption. Alcoholic fatty liver is usually characterized by liver enlargement, increased serum and liver TG levels, and many fat droplets in sections of the liver tissue [27]. The response to alcohol intake depends on underlying hypertriglyceridemia and hypercholesterolemia, which lead to increased TG and TC accumulation in the liver [20]. Our study results show that the serum levels of TC and TG were significantly increased by alcohol administration, but the treatment with FGK decreased those serum lipid profiles (Figure 5). In addition, our histological evaluation of murine liver tissue showed that the FGK decreased lipid accumulation (Figure 2). These findings suggest that FGK effectively protects against alcohol-induced fatty liver injury.

Alcohol intake causes liver inflammation associated with oxidative stress, and that inflammatory response leads to an overproduction of inflammatory cytokines such as TNF-α, IL-1β, and IL-6 [15]. In addition, pro-inflammatory mediators such as iNOS and COX-2 perform a major role in the liver damage caused by alcohol administration, and iNOS activation can induce high levels of NO production [9]. Our results confirm that the FGK effectively inhibited the production of inflammatory cytokines in the sera and livers of alcohol-treated mice (Figures 6 and 7). The increased expression of Th1-type cytokines such as TNF-α contributes to alcohol-induced fatty liver disease [28], and FGK administration decreased the expression of TNF-α and thereby relieved the inflammatory liver damage caused by alcohol in the body (Figures 6a and 7d). In addition, our results show that the FGK suppressed the overexpression of iNOS and COX-2 in the livers of alcohol-treated mice (Figure 7b,c). These results indicate that a pretreatment with FGK can downregulate liver inflammation and thereby prevent alcoholic liver damage.

Several studies suggest that increased inflammatory responses from alcohol consumption play an important role in developing liver disease [29]. Alcohol administration provides an inflammatory environment in the liver by activating inflammatory macrophages [30,31]. In this study, our results confirmed that increased inflammatory mediators in the blood and liver of alcohol-treated mice are helpful to the normal condition

range when treated by FGK. Thus, FGK may have beneficial functional food materials in the treatment of ALD.

In summary, liver damage induced by the consumption of alcohol is associated with an inflammatory response via enhanced lipid peroxidation and alcohol metabolism. Our results show that FGK hepatoprotects the effects against alcohol-induced liver damage by having anti-inflammatory effects, inhibiting lipid profiles, and controlling alcohol metabolism. These findings suggest that FGK could be developed into a beneficial functional food to protect against alcohol-induced liver injury. In addition, these results provide mechanistic insight into the hepatoprotective effects of FGK against alcoholic liver disease. Further studies are needed to investigate the molecular mechanism research in the ALD model for its utilization as a promising protective agent.

### 5. Conclusions

In conclusion, treatments with fermented gold kiwi could effectively alleviate chronic alcohol-induced liver damage in mice by regulated alcohol-metabolism (ethanol, aldehyde, ALD, and ALDH), anti-inflammation (iNOS, COX-2, TNF-$\alpha$, IL-1$\beta$, and IL-6), and declined lipid profile levels (TC and TG). Our study demonstrated that a fermented gold kiwi treatment could reduce the alcoholic liver damage response and also provides a reference for the future research and treatment of ALD.

**Author Contributions:** Conceptualization, H.K. and J.K.; methodology, J.C. and J.K.; validation, J.C., S.L., H.C. and J.L.; resources, N.L., H.O. and H.K.; data curation, J.C.; writing—original draft preparation, J.C.; writing—review and editing, J.C. and J.K.; project administration, H.K. and J.K. All authors have read and agreed to the published version of the manuscript.

**Funding:** This research was financially supported by the Ministry of Small and Medium-sized Enterprises (SMEs) and Startups (MSS), Korea, under the "Regional Specialized Industry Development Plus Program (R&D, S3260875)" supervised by the Korea Institute for Advancement of Technology (KIAT).

**Institutional Review Board Statement:** This study was approved by the Jeonbuk National University Institutional Animal Care and Use Committee (JBNU 2022-095) and conducted according to the guidelines for animal experiments in the protocol.

**Informed Consent Statement:** Not applicable.

**Data Availability Statement:** Not applicable.

**Acknowledgments:** This research was supported by Vitech Co., Ltd.

**Conflicts of Interest:** The authors declare no conflict of interest.

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
