# Peer review of "Fermented Gold Kiwi Prevents and Attenuates Chronic Alcohol-Induced Liver Injury in Mice via Suppression of Inflammatory Responses"

_applsci, doi:10.3390/app13031877_

Round 1

Reviewer 1 Report

The results part was well presented, but needs more discussion in this research and supported by enough updated references.

The author needs to strictly revise the high similarity index, which has a similarity of 10% and 7% with primary sources.

The authors need to strictly use the updated references; less than 50% of the references cited in this manuscript were out of date (more than five years ago)

There is no conclusion part in this manuscript.

Author Response

On behalf of the authors, I would like to thank you for providing us the opportunity to improve our manuscript once again. We appreciated the careful reading of our manuscript as well as commenting and suggesting for better manuscript.

We have carefully rewritten and reorganized our manuscript according to the comments from you.

We hope that you agree with our manuscript that has been not only through revised but also strengthened by your comments.

Thank you for your kind consideration.

The results part was well presented, but needs more discussion in the research and supported by enough updated references

Response: We added more discussion part. Thank you for your consideration.

The author needs to strictly revise the high similarity index, which has a similarity of 10% and 7% primary sources.

Response: The number 1 and 2 are "mpdi" in the report you sent me is ambiguous, so I used a plagiarism program called "copykiller" to conduct further confirmation. No. 1 reference is our previous study; this material (fermented gold kiwi) and recent study material use the same sample preparation method. Therefore, please consider that it can show some similarity. (This paper is a follow-up to the previous study.)

The authors need to strictly use the updated references; less than 50% of the references cited in this manuscript were out of date (more than five years ago)

Response: We revised the part according to the reviewer's opinion and the reference. Thank you for your careful review.

There is no conclusion part in this manuscript

Response: We added the conclusion part in this manuscript. Thank for your consideration.

Reviewer 2 Report

Dear Authors:

This study investigates the possible hepatoprotective effects of FGK in alcohol-induced liver injury by detecting alcohol metabolism and inflammatory markers in mice. However, I have some questions before go-ahead:

Why the authors did measurement lipid profile even if it is unspecified during alcoholic liver disease? Why did not do a qPCR to evaluate the ALD, ALDH and CYP2E1 activities?

English editing services should be acquired to improve mistakes in:

Introduction:

2nd, 4th, 5th paragraphs

Material and methods:

2.4. Biochemical Assay Measurement 

2.5. Histological Analysis

2.6. Western Blotting

2.7. Statistical Analysis

Results:

3.4. Effects of Fermented Gold Kiwi Powder On Alcohol-Induced Inflammatory Mediators

1st, 2nd and 3rd paragraphs

Discussion:

1st and 4th paragraphs.

Kind regards, 

Author Response

On behalf of the authors, I would like to thank you for providing us the opportunity to improve our manuscript once again. We appreciated the careful reading of our manuscript as well as commenting and suggesting for better manuscript.

We hope that you agree with our manuscript that has been not only through revised but also strengthened by your comments.

Thank you for your kind consideration

Dear Authors:

This study investigates the possible hepatoprotective effects of FGK in alcohol-induced liver injury by detecting alcohol metabolism and inflammatory markers in mice. However I have some question before go-ahead:

Why the authors did measurement lipid profile even if it is unspecified during alcoholic liver disease? Why did not do qPCR to evaluate the ALD, ALDH and CYP2E1 activities?

Response: We extracted proteins from liver tissue in mice with alcoholic liver damage and performed western blots on alcohol dehydrogenase (ADH) and aldehyde dehydrogenase (ALDH). The results of liver protein expression are shown in Figure 4, so we did not evaluate the qPCR.

English editing services should be acquired to improve mistakes in:

Introduction: 2nd, 4th, 5th paragraphs

Material and methods: 2.4. Biochemical Assay Measurement, 2.5. Histological Analysis, 2.6. Western Blotting, 2.7. Statistical Analysis

Results: 3.4. Effects of Fermented Gold Kiwi Powder On Alcohol-Induced Inflammatory Mediators,1st, 2nd and 3rd paragraphs

Discussion: 1st and 4th paragraphs.

Response: We already have asked an English editing center “eWorldEditing” to our manuscript before submission at applied sciences. We attached a certificate that we received English editing services for this manuscript.

Round 2

Reviewer 1 Report

The authors have revised the manuscript as requested

The manuscript needs grammar proof before the publishing process

Author Response

Thank you for your consideration. We asked the editor about the English grammar editing for this manuscript. The English grammar editing will also be performed by “MDPI” English editors if manuscript is accepted.

Reviewer 2 Report

Dear authors:

The manuscript still flaws in English editing. Even if a certification has been added, other professional views are needed.

Kind regards,

Author Response

(The authors gave the same response as above.)
